# Semi-Supervised Classification and Landscape Metrics for Mapping and Spatial Pattern Change Analysis of Tropical Forest Types in Thua Thien Hue Province, Vietnam

**Truong Thi Cat Tuong [1,2]**, **Hiroshi Tani [3,\*]**, **Xiufeng Wang [3]** and **Nguyen Quang Thang [4]**

[1] Mientrung Institute for Scientific Research, Thua Thien Hue Province 530000, Vietnam
[2] Graduate School of Agriculture, Hokkaido University, Sapporo 060-8589, Japan
[3] Research Faculty of Agriculture, Hokkaido University, Sapporo 060-8589, Japan
[4] Central Sub Forest Inventory and Planning Institute, Thua Thien Hue Province 530000, Vietnam
\* Correspondence: tani@env.agr.hokudai.ac.jp; Tel.: +81-011-706-3659

**Abstract:** Research Highlights: In this study, we classified natural forest into four forest types using time-series multi-source remotely sensed data through a proposed semi-supervised model developed and validated for mapping forest types and assessing forest transition in Vietnam. Background and Objectives: Data on current forest state and changes detection are always essential for forest management and planning. There is, therefore, a need for improved tools to classify and evaluate forest dynamics more accurately and effectively. Our objective is to develop such tools using a semi-supervised model and landscape metrics to classify and map changes in natural forest types by using multi-source remotely sensed data. Materials and Methods: A combination of Landsat data with PALSAR and PALSAR-2 was used for forest classification through the proposed semi-supervised model. This model turned a kernel least square into a self-learning algorithm, trained by a small number of samples with given labels, and then used this classifier to assign labels to the unlabeled data. The overall accuracy, kappa, user's accuracy, and producer's accuracy were used to evaluate the classification accuracy by comparing the classified image with the results of ground truth interpretation. Based on the classified images, forest transition was evaluated using certain landscape metrics at the class and landscape levels. Results: The multi-source data approach achieved improved discrimination of forest types compared to only using single data (optical or radar data). Good classification accuracies were obtained, with kappas of 0.81, 0.76, and 0.74 for the years 2007, 2010, and 2016, respectively. The analysis of landscape metrics indicated that there were different behaviors in the four forest types, as well as provided much information about the trends in spatial pattern changes. Conclusions: This study highlights the utilization of a semi-supervised model in forest classification, and the analysis of forest transition using landscape metrics. However, future research should include a comparison of different models to estimate the improvement of the proposed model. Another important study that should be conducted is to test the proposed method on larger areas.

**Keywords:** forest types classification; forest transition; semi-supervised model; landscape metrics; Landsat data; synthetic aperture radar

## 1. Introduction

Since the early 1990s, the tropical forest in several countries has been undergoing a transition period from degradation to reforestation [1–3]. Forest transition is considered from the perspective of forest area changes and the conversion from other land use/land cover types to forest. With the

rapid development of remote sensing technology and the wide application of landscape ecology, they supply effective tools to analyze spatial-temporal changes and related ecological processes. Improved understanding of forest transition provides many benefits, such as global carbon balance or land use and forest policy implementation [4,5]. Therefore, there is a need to further develop new methods for forest type classification and forest transition assessment.

Recently, remote sensing combined with the conventional method to supply validation data has been extensively used in forest inventory. The advantages of the remote sensing technique are cost- and labor-saving as well as swift observation of large scale forest changes over the long term. However, the classification accuracy associated with using remote sensing is affected by many factors, such as the classification techniques, training samples, and the signal reflected from objects.

A natural forest [6] is a naturally regenerated forest comprising native species, where there are no clear or clearly visible indications of human activities and the ecological processes are not significantly disturbed. In this study, we classified natural forests based on the timber reserve of standing trees into four main types: rich, medium, poor, and restoration forest. Although these four types differ in species composition and timber reserves, we found that with only a single source of data (optical or radar data) it is often difficult to discriminate between different kinds of natural forest types because of the very similar information on canopy and forest structure captured by remotely sensed data [7]. This highlights the need for multi-source remote sensing data to extract more information of interest regarding the objects for classification. By using multi-source data, the classification accuracy is improved compared to single data source. This has been shown, for example, with a combination of optical data and synthetic aperture radar (SAR, Congo Basin and Malawi city, Mzimba) [8,9]. The fusion of different frequencies (L– and P–band) of SAR products has also received much attention in recent years [10–12].

Another challenge is that sampling is restricted because of the complexity of ecosystems and inaccessible regions [13]. In this study, we used semi-supervised classification to overcome the paucity of ground truth samples. Semi-supervised classification focuses on enhancing supervised classification by minimizing errors in the labeled examples, but it must also be compatible with the input distribution of unlabeled instances [14]. While supervision often provides higher classification accuracy, it requires a good dataset to ensure both the quantity and quality of training samples collected from the field survey. The constraint of field data collection is that it is not always achievable, owing to limitations in finance, terrain, or availability of the data source. To avoid this issue, semi-supervised classification aims at solving the limited number of labeled samples and taking advantage of the abundant unlabeled samples. Many semi-supervised classification algorithms such as expectation-maximization, co-training, and self-training have been developed. The graph-based method has also attracted an increasing amount of interest [15–18]. This method works by summarizing base model outputs in a group-object bipartite graph and maximizing the consensus by promoting smoothness of label assignment over the graph and consistency with the initial labeling. Recently, machine learning has received much attention and has been applied to the semi-supervised learning problem. This technology has been successfully developed for binary classification, such as in [19], where a Laplacian Twin Support Vector Machine was used for semi-supervised classification that can exploit the geometry information of the marginal distribution embedded in unlabeled data to construct a more reasonable classifier-semi-supervised classification with graph convolutional networks [20] which scales linearly in the number of graph edges and learns hidden layer representations that encode both the local graph structure and the features of nodes.

For land use/land cover, semi-supervised classification has been successfully adopted in the literature. For instance, in [21], semi-supervised logistic regression was applied. This is a specific instance of the generalized maximum entropy that finds a probability distribution that minimizes a divergence based on the entropy of the weights of classifiers. In [22], a semi-supervised clustering was presented that is simultaneously optimized using a modern multi-objective optimization technique based on the concepts of simulated annealing. In [23], the weight support vector machine was used

to keep the training effort low with a manually-collected set of pixels of the class of interest and a random sample of pixels. In [24], extended label propagation and rolling guidance filtering that uses superpixel propagation were applied to assign the same labels to all pixels within the superpixels that are generated by the image segmentation method.

In this paper, we present a self-learning approach for forest classification that can propagate labels from labeled samples to unlabeled data to build a large volume of training data. This model does not make any specific assumptions for the input data, but it does accept that its own predictions tend to be correct [14]. Self-learning, also known as Yarowsky's algorithm, is a rule-based semi-supervised classification. The term "self-learning" is used because the algorithm uses its own prediction to teach itself. Self-learning is very popular, with an initial classifier trained by a small number of training data with given labels, before using this classifier to assign labels to the unlabeled sample. For each unlabeled sample, confidence values are extracted from the probabilistic of learning models [14,25]. The samples that have been labeled with the most confident prediction are then selected to combine with the training data and create a new training set. The classifier is then retrained on that new training set and the procedure repeated. Self-learning has been applied in several text processing tasks in the last few years. Recently, it has been applied with some developed supervisor classifiers to image classification [23,26]. This study developed self-learning with a kernel least square classifier for forest types classification. Least squares is a standard approach of statistical analysis and has been well-known for a long time. It was developed by applying kernel functions in high dimensional feature space to solve the problem of a large number of parameters [27]. Kernel functions are an algorithm with the advantage of being able to flexibly transform an originally non-linear vector into a linear version in feature space. Therefore, they are widely applied in solving classification problems involving multiple features [28–30].

In this study, we also used time-series remotely sensed data for the evaluation of forest changes by landscape ecology. Landscape ecology can be generally defined as the science and art of studying and improving the relationship between spatial patterns and ecological processes on a multitude of scales and organizational levels [31]. One fundamental aspect has been its explicit attention to the spatial dimension of ecological processes [32]. Landscape metrics are one of the classical landscape ecological tools for measurement, analysis, and interpretation of spatial patterns [33]. The contribution of remote sensing to landscape planning and conservation is mainly in the inventory and determination of objects of interest and in monitoring changes by time-series satellite data [34]. A basic concern in forest management is spatial processes over time, such as deforestation, degradation, or restoration. The analysis of landscape structure is a classic approach for the understanding of spatial processes using various landscape metrics [32,35–37]. Several studies provide evidence of the value of remote sensing and landscape metrics for forest management [38–42].

In summary, there are two main objectives in this study. The first objective is to assess the potential of a semi-supervised model to classify natural forest types by using multi-source remote sensing data. The second objective is to assess the process of forest transition from the perspective of landscape ecology by using multi-temporal data.

## 2. Study Area

In Vietnam, the forest plays an important role in the socio-economic system in the mountainous province, where local people have a low income and agroforestry-based livelihoods. Although centralization of forest resource management began in Vietnam very early in the 1950s [4], the natural forest experienced a rapid decrease over the long term [43], causing negative impacts to the environment, such as loss of carbon stock, biodiversity degradation, and habitat fragmentation [44]. Since 2005, however, Vietnam has been experiencing a positive period in the application of forestry policies [45], which is contributing to development of the forested area. This dramatic forest transition has resulted in changes to the biophysical, ecological process, as well as to the spatial landscape. However, there is a lack of up-to-date information on forest changes in Vietnam in the period from 2005 to the present,

particularly in central Vietnam where the socio-economic dynamics have recently been increasing. To create a reliable forest management strategy, an improved understanding of forest changes is essential. This can be achieved by spatial analysis through multi-temporal remote sensing images processing, combined with landscape metrics assessment.

Thua Thien Hue province, located in central Vietnam (Figure 1d), has a surface area of 5054 km$^2$ and the natural forest area accounts for approximately 40% of the total area. According to the General Statistics Office (GSO) in Vietnam, the natural forest in this study area slightly decreased from 203,800 ha in 2008 to 202,700 ha in 2010, with the principal causes of deforestation comprising the conversion from forest to other land uses (e.g., hydropower, roads, cultivation) and illegal exploitation of forest products. Conversely, from 2010, there was a significant extension of natural forest with the area reaching 212,200 ha in 2016. These fluctuations have not only caused changes in the area, but also in the forest landscape structure.

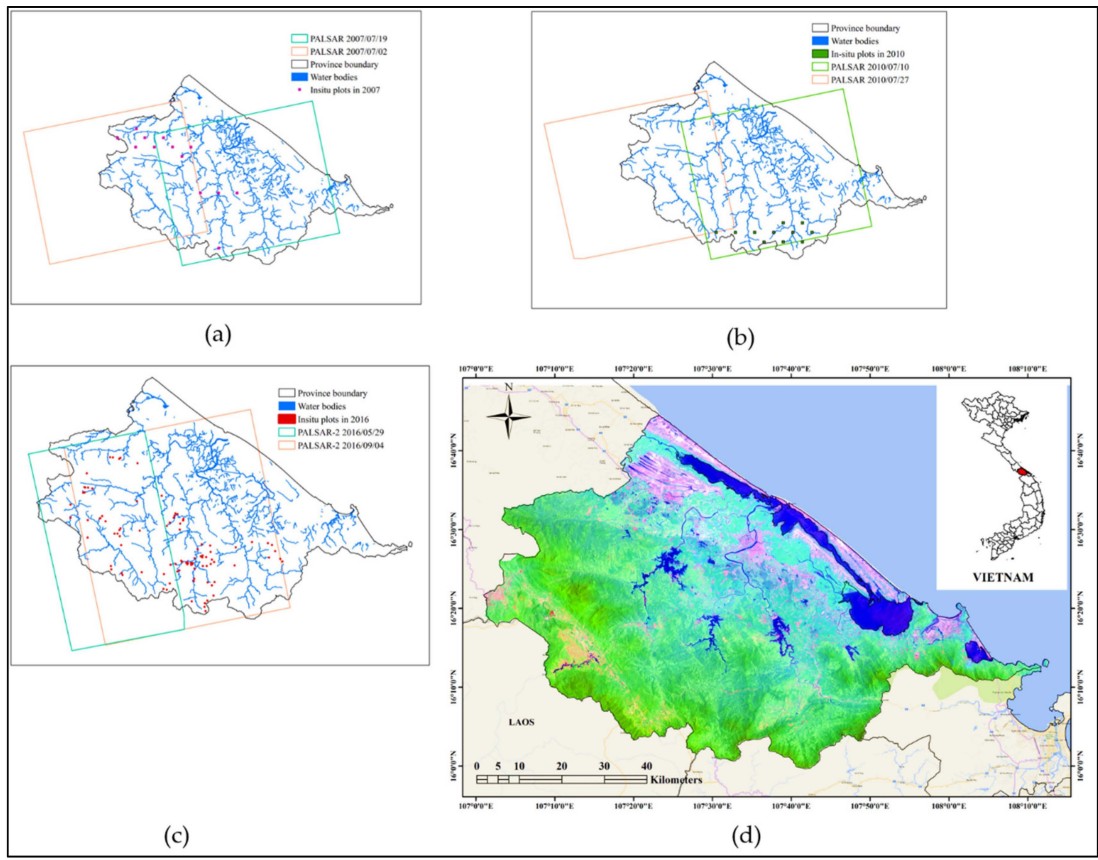

**Figure 1.** Cover of synthetic aperture radar (SAR) images and in-situ data in (**a**) 2007, (**b**) 2010, (**c**) 2016, and (**d**) location map of the study area in Landsat data with pseudo colors (R: SWIR 2, G: near-infrared, B: green).

We classified the natural forest into four types based on the specific condition of the study site as well as circular number 34/2009/TT-BNNPTNT of June 10, 2009 [46] published by Vietnam Ministry of Agriculture and Rural Development, on the criteria for forest identification and classification in Vietnam:

1.  Rich forests are forests with a timber reserve of standing trees of between 201 and 300 m$^3$/hectare;
2.  Average forests (or medium forests) are forests with a timber reserve of standing trees of between 101 and 200 m$^3$/hectare;
3.  Poor forests are forests with a reserve of standing trees of between 10 and 100 m$^3$/hectare;

4. Forests with no reserve ("Restoration forest" in the case of our study site) are forests with a timber tree average diameter of less than 8 cm and a timber reserve of standing trees of less than 10 m$^3$/hectare.

## 3. Data and Methods

### 3.1. Data

We used time-series SAR data and Landsat data acquired in 2007, 2010, and 2016 (Figure 1a,b,c). Two scenes of SAR data were collected per year, which were then used to create a mosaic covering 77% of the study area. The SAR data differed in term of acquisition mode, which led to a difference in the incidence angle and the size of the range and azimuth. Therefore, preprocessing was necessary to synchronize these data. Two polarization HH (horizontal transmitting, horizontal receiving) and HV (horizontal transmitting, vertical receiving) were used to process the data in this study. Landsat data were also selected to combine with SAR data for forest type classification. Landsat data were provided by the United States Geological Survey (USGS) with moderate resolution and wide spectral coverage. The swath width of Landsat is 185 km; therefore, it could cover the full study area. The characteristics of these data are described in Table 1.

**Table 1.** Characteristics of satellite image data used in this study.

| Date | Types | Level | Incidence Angle at Scene Center | Resolution (m) | Polarization/Band |
|------|-------|-------|--------------------------------|----------------|-------------------|
| 2016/05/29 | PALSAR2 | 1.1 | 38.99 | 3.12 × 4.55 | HH + HV + VH + VV |
| 2016/09/04 | PALSAR2 | 1.1 | 40.5 | 3.4 × 6.6 | HH + HV |
| 2010/07/10 | PALSAR | 1.1 | 38.7 | 3.2 × 15 | HH + HV |
| 2010/07/27 | PALSAR | 1.1 | 38.7 | 3.2 × 15 | HH + HV |
| 2007/07/02 | PALSAR | 1.5 | 38.7 | 12.5 | HH + HV |
| 2007/07/19 | PALSAR | 1.5 | 38.7 | 12.5 | HH + HV |
| 2007/04/24 | Landsat TM | 1 | - | 30 | 5 |
| 2010/02/11 | Landsat TM | 1 | - | 30 | 5 |
| 2016/04/16 | Landsat OLI | 1 | - | 15, 30 | 11 |

A ground sample was also collected to support training data and accuracy assessment. These data were provided by the Central Sub Forest Inventory and Planning Institute, Thua Thien Hue province, Vietnam (Sub-FIPI). The data collection was evenly distributed over the entire study area at three time periods—In 2007, 2010, and 2016. The samples were then divided into 80% training data and 20% validation data. In 2007, 13 measured plots were covered by the PALSAR scene, with each plot measuring 1 km$^2$ (1000 × 1000 m), while in 2010, there were 10 such plots. In each plot, 40 subplots of 25 × 20 m were set to measure forest parameters and describe characteristics. However, not all 40 subplots were measured and selected for classification; only some met the conditions of being natural forests with reserves, not separated by other obstacles such as rivers, streams and roads, and terrain. In 2007, 170 subplots were selected for this study, while in 2010, 115 subplots were selected. In 2016, 106 plots were covered by PALSAR-2 data. Each rectangular plot measured 30 × 33 m with the longer aspect running in an east-west direction and the shorter aspect running north-south. The distribution of samples for the four forest types is described in Table 2.

**Table 2.** Ground data for the four forest types in the study area in 2007, 2010, and 2016.

| Types | Number of Samples | | |
|---|---|---|---|
| | **2007** | **2010** | **2016** |
| Rich forest | 17 | 20 | 29 |
| Medium forest | 68 | 34 | 23 |
| Poor forest | 48 | 34 | 37 |
| Restoration forest | 37 | 27 | 17 |
| Total | 170 | 115 | 106 |

Apart from these samples, a larger amount of unlabeled data was supplied for forest types classification. A total of 200 unlabeled samples was randomly created over the study area. The proportion of unlabeled samples accounted for approximately 40–60% of the total samples to ensure the accuracy of the classification results. In particular, the number of unlabeled samples was equivalent to 55% for 2007, 64% for 2010, and 65% for 2016.

*3.2. Methods*

A flowchart of the methodology employed in this study is presented in Figure 2.

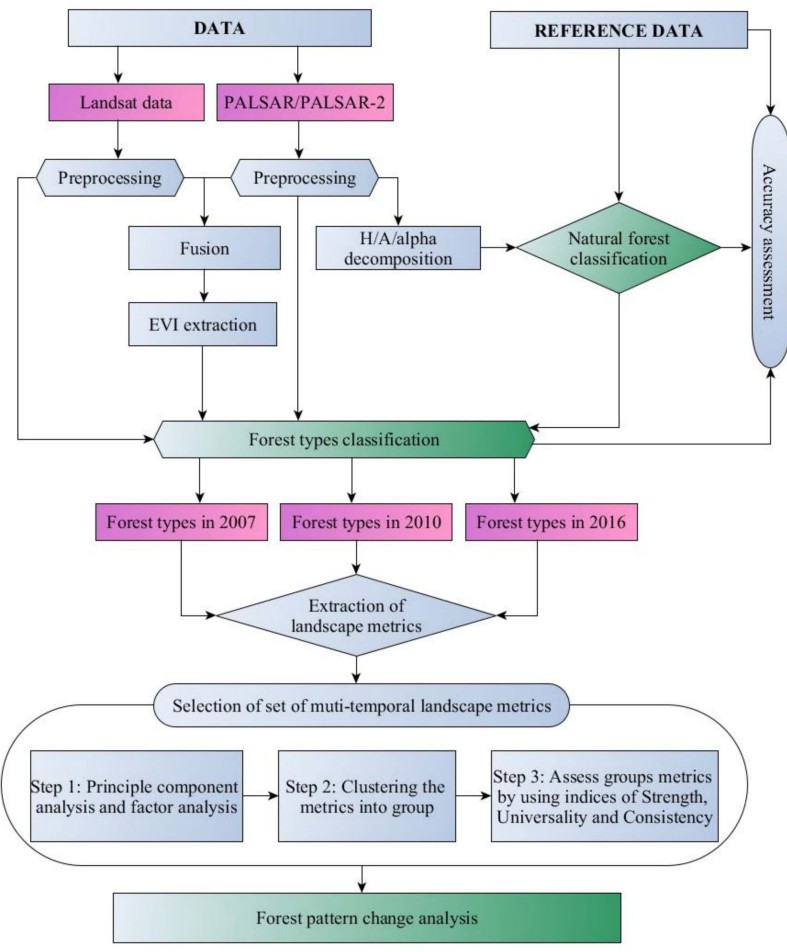

**Figure 2.** Flowchart of the methodology employed in this study.

3.2.1. Preprocessing

Landsat digital numbers (DNs) were converted to reflectance and atmospheric correction using the fast line-of-sight atmospheric analysis of hypercubes (FLAASH) tool. The enhanced vegetation

index (EVI) was then calculated using band near-infrared (0.7–1.1μm), red (0.6–0.7 μm), and blue (0.45–0.52 μm) in accordance with the work of Liu and Huete (1997) [47]:

$$EVI = G \times \frac{\rho_{nir} - \rho_{red}}{\rho_{nir} + (C_1 \times \rho_{red} - C_2 \times \rho_{blue}) + L} \qquad (1)$$

where $L$ is a soil adjustment factor, and $C_1$ and $C_2$ are coefficients used to correct aerosol scattering in the red band by using the blue band. In general, $G = 2.5$, $C_1 = 6.0$, $C_2 = 7.5$, and $L = 1$.

In this study, when observing the relationship between reflectance value and the cosine of the solar incidence angle, there was a low correlation coefficient with the value of 0.0075 and 0.0197 for TM and OLI data, respectively. This means that the terrain does not significantly affect this test site. Therefore, topography correction is unnecessary in this case.

For radar data, dual-polarized images (HH, HV polarizations) were created in the single-look complex (SLC) format. The preprocessing data were operated to convert the digital number value into sigma naught ($\sigma^o$) values using the following equation:

$$\sigma^o = 10.\log_{10}(I^2 + Q^2) + CF - A \qquad (2)$$

where $I$ and $Q$ are the real and imaginary parts of the SLC product. A is a conversion factor equal to 32.0. The calibration factor CF is -83.

A refined Lee filter was used with a window size of $7 \times 7$ to reduce the speckle noise. The topography effect was eliminated using range—Doppler terrain correction with digital elevation model (DEM) from the Shuttle Radar Topography Mission, and all of the product images were resampled to reach 15 m in pixel spacing.

The preprocessed SAR data were next transformed into covariance matrix elements, and then eigenvalue and eigenvector polarimetric parameters. The cross-pol ratio of HH and HV was also calculated and used as a variable for the classification model. In addition, SAR data and Landsat data were fused and resampled to 15 m. The parameters set for polarimetric SAR (PolSAR) and Landsat data comprise the input features for classification, as detailed in the next section.

To illustrate the polarimetric data, we adopted eigen decomposition of the $2 \times 2$ covariance matrix for dual polarization data as defined by [48]:

$$\begin{bmatrix} C_{HH,HH} & C_{HH,HV} \\ C_{HV,HH} & C_{HV,HV} \end{bmatrix}$$

H/A/Alpha decomposition was used to decompose the backscatter value into three components: entropy, anisotropy, and alpha (H/A/$\alpha$). The H/A/$\alpha$ is a polarimetric parameters decomposition based on eigenvalue and eigenvector that was introduced by Cloude and Pottier [49]. In this technique, backscattering is decomposed into entropy (H), anisotropy (A), and alpha angle ($\alpha$). Entropy is a parameter describing randomness in target scattering, which is defined as:

$$H = -\left(\overline{\lambda}_1 ln\overline{\lambda}_1 + \overline{\lambda}_2 ln\overline{\lambda}_2\right)/ln2 \text{ with } \overline{\lambda}_i = \lambda_i/(\lambda_1 + \lambda_2) \qquad (3)$$

where $H_T$ is target entropy and $\lambda_i$ (i = 1 to 2) are eigenvalues.

Entropy values vary from 0 for a single scattering mechanism to 1 for pure noise and random targets.

Mean alpha angle is defined as:

$$\alpha = \overline{\lambda}_1 \alpha_1 + \overline{\lambda}_2 \alpha_2 \qquad (4)$$

The alpha angle varies between 0° for trihedral scattering from a planar surface to 90° for dihedral scattering from a metallic surface. Another element is anisotropy (A), which is a parameter complementary to entropy, which can be employed as a source of discrimination only when H >0.7 owing to the high effect of noise [50].

### 3.2.2. Masking Undesirable Areas

In this study, we created a mask to remove undesirable areas before classifying natural forest types. The classification method of the random forest algorithm was applied based on entropy, alpha, and anisotropy parameters extracted from dual polarization data for images in 2010 and 2016. For the image in 2007, the polarization data of HH, HV, and EVI from Landsat data were used for classification. For other land use/land cover types, samples such as rivers, urban areas, and agricultural land were collected through visual interpretation based on discrimination in color, geometric shapes, and brightness. For the natural forest, 170 samples were collected for 2007 with a plot area of $25 \times 40$ m, 115 samples for 2010, and 106 samples for 2016 with the same area of $30 \times 33$ m. Polarimetric data were derived from the image for each sample with a window size of $2 \times 2$ pixels, with a pixel size of 15 m. The classification results create natural forest maps for the study area.

Furthermore, in this study area, because the natural forest is mainly distributed on topography at an elevation above 200 m, a digital elevation map (DEM) was applied to mask out low-altitude forest areas while retaining forests with elevations above 200 m. This DEM map was downloaded from NASA Shuttle Radar Topography Mission data. The masked forest images were then used for the forest types classification.

### 3.2.3. Self-Learning with the Kernel Least Squares (SL-KLS) Classifier for Forest Types Classification

Kernel Least Squares (KLS)

In this study, the presence of a large number of parameters in the classification problem created computational difficulties due to a high number of dimensions. To solve this problem, we used the KLS technique in the R environment with RSSL package version 0.7. Here, KLS is described as a method using least squares regression as a classification technique with numeric encoding of classes as targets. A detailed description of KLS can be found in various studies [27,51], with the optimal parameter vector identified by $\theta = [b\alpha_1\alpha_2 \dots \alpha_n]^T$. The minimized vector has the form $L(\theta) = \|Y - P\theta\|^2$,

$$\text{with } Y = [y_1 y_2 \dots y_n]^T \text{ and } P = \begin{bmatrix} 1 & k[x_1, x_2] & \cdots & k[x_1, x_n] \\ \vdots & \vdots & \ddots & \vdots \\ 1 & k[x_n, x_1] & \cdots & k[x_n, x_n] \end{bmatrix}$$

A radial basis function was used with the form below:

$$k(x_i, x_j) = \exp\left(-\frac{\|x_i - x_j\|^2}{\sigma^2}\right) \tag{5}$$

where $x_i$: are training data, $x_j$ is a feature vector, and $\sigma$ is a free parameter. Kernel k has a value in the range of 0 to 1. With $\alpha_i$ as real numbers, the prediction function $f(x)$ can be written as follows:

$$f(x) = \sum_{i=1}^{n} \alpha_i k(x_i, x) + b \tag{6}$$

Self-learning with the Kernel Least Squares (SL-KLS) Classifier

In this study, a self-learning algorithm was used to turn the KLS classifier into a semi-supervised model to solve the problem of the small amount of labeled data. Based on the training data, KLS was applied to assign labels to unlabeled objects, which were then added to the set of labeled objects for classification. There is a given set of labeled data (L) and a set of unlabeled data (U) (Figure 3). By applying a KLS classifier, k number of labels are assigned to unlabeled data. The result of predicted data U then joins with L to create a new training set for classifying the entire segmented images. In this study, we classified the forest into four classes: rich forest, medium forest, poor forest, and

restoration forest. The features of the four classes were extracted from Landsat bands reflectance, EVI, HH, HV signals, covariance elements, and H/A/Alpha decomposition.

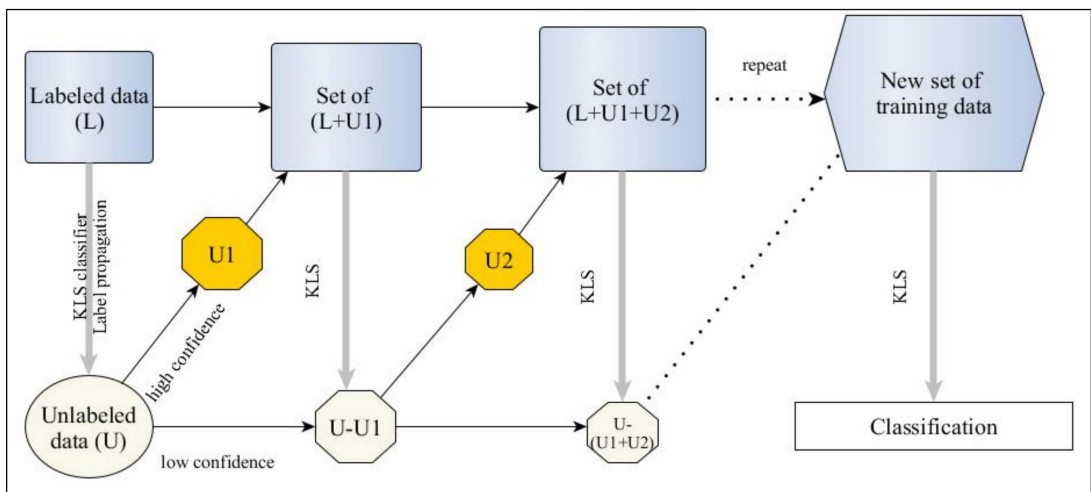

**Figure 3.** Flowchart of classification using the combination of self-learning with kernel least squares classifier in this study.

The indicator of overall accuracy (OA), kappa, user's accuracy, and producer's accuracy were used to evaluate classification accuracy by comparing the classified image with the results of ground truth interpretation. The overall accuracy comprises the ratio of the sum of accuracy in an individual class and the number of observed samples, with 100% as the perfect classification. Kappa, user's, and producer's accuracy were proposed by Congalton and have been used widely to date. The function of these indicators is clearly described in [52].

### 3.2.4. Forest Pattern Analysis Using Landscape Metrics

Extraction of Landscape Metrics

After the classification step, the forest was divided into four forest types: rich, medium, poor, and restoration forest in the years 2007, 2010, and 2016. For each year, the classified images were then clipped into 14 non-overlapping sub-landscapes of 2000 × 2000 m. This size was selected to ensure the representativeness of the sample and to reduce computation time. To conduct spatial analysis of the forest pattern, landscape metrics were computed at two levels, class and landscape, for all samples in each year. We calculated 56 metrics at the class level and 63 metrics (Appendix A) at the landscape level for each sub-landscape image using Fragstat version 4.2.1. With a large number of landscape metrics, we then selected the appropriate metrics for analysis of the natural forest process for the study area longitudinally.

Selection of a Set of Landscape Metrics

Principal components analysis was used to identify components and cluster them into various groups. In these groups, the three indices of universality, consistency, and strength were then calculated to select the group of metrics. This operation was conducted using PROC FACTOR in SAS.

Based on the assessment of metrics through the three indices of universality, consistency, and strength, we created a list of selected metrics at the class and landscape levels. At the class level, 11 clusters were created from 56 metrics. Through cluster analysis, two clusters (approximately 16 metrics) were selected with a high level of these three indices at a total percentage >90%, variation explained >7%, and the average in-group correlation >0.8 (Appendix B). Similarly, two clusters (approximately 20 metrics) were selected through analysis of the 10 clusters created from 63 metrics for

the landscape level (Table 3). The other two metrics—total area (CA in hectares_ha) and percentage of landscape (PLAND_%)—were also added for change analysis of the area in general.

**Table 3.** Set of high representative metrics for analyzing multi-temporal forest types structure at class and landscape level in the study area.

| No | | Metric Name | Level | Description |
|----|----|----|----|----|
| 1 | | AI | C | Aggregation index |
| 2 | | CLUMPY | C | Clumpiness index |
| 3 | Aggregation/ Fragmentation | COHESION | C | Patch cohesion |
| 4 | | NLSI | C | Normalized landscape shape index |
| 5 | | PLADJ | C | Proportion of like adjacencies |
| 6 | | IJI | L | Interspersion/ juxtaposition index |
| 7 | | MESH | L | Effective mesh size |
| 8 | Area and edge metrics | AREA_AM | L | Area-weighted mean patch size |
| 9 | | AREA_CV | L | Patch size coefficient of variation |
| 10 | | GYRATE_AM | L | Area-weighted radius of gyration |
| 11 | | CAI_CV | C | Core area coefficient of variation |
| 12 | | CORE_AM | L | Area-weighted mean core area |
| 13 | Core area metrics | CORE_CV | L | Core area coefficient of variation |
| 14 | | DCORE_AM | L | Area-weighted mean disjunct core area |
| 15 | | DCORE_CV | L | Disjunct core area coefficient of variation |
| 16 | | CIRCLE_AM | C | Area-weighted related circumscribing circle |
| 17 | | CIRCLE_CV | C, L | Circumscribing circle coefficient of variation |
| 18 | | CIRCLE_MN | C, L | Mean related circumscribing circle |
| 19 | | CONTIG_AM | C | Area-weighted contiguity index |
| 20 | | CONTIG_MN | C, L | Mean contiguity index |
| 21 | | CONTIG_CV | L | Contiguity index coefficient of variation |
| 22 | Shape metrics | SHAPE_MN | C, L | Mean shape index |
| 23 | | SHAPE_AM | L | Area-weighted mean shape index |
| 24 | | SHAPE_CV | L | Shape index coefficient of variation |
| 25 | | FRAC_AM | L | Area-weighted mean fractal dimension |
| 26 | | FRAC_MN | C, L | Mean fractal dimension |
| 27 | | FRAC_CV | C, L | Fractal dimension coefficient of variation |
| 28 | | PARA_MN | C, L | Mean perimeter–area ratio |
| 29 | | PARA_AM | C | Area-weighted mean perimeter–area index |

Analysis of Forest Pattern Change

From the set of representative metrics in the study area, we selected various metrics that support the analysis of spatial processes over time, containing aggregation, compactness, and fragmentation. To evaluate the spatial structure change of forest types in the period 2007–2016, we selected 11 metrics for class level and five metrics for landscape level.

The aggregation is expressed by increasing the size of patches from the combination of small fragments. Therefore, this indicator relates to the recovery of forests from the previously deforested area. The metrics are related to aggregation including the aggregation index (AI), proportion of like adjacencies (PLADJ), and clumpiness index (CLUMPY) for the class level, and Interspersion/juxtaposition index (IJI) for the landscape level. Another term that is strongly involved in the aggregation is forest connectivity, which evidently increases the patch cohesion index (COHESION) that is related to the physical connectedness of the corresponding patch type.

Forest fragmentation is an opposite process to aggregation and occurs when a large contiguous forest is broken down into many small fragments, leading to loss of biodiversity and animal habitat and degradation of forest health and its economic and environmental functions. This process is closely related to the shrinkage ratio of area-weighted mean patch size (AREA_AM) and effective mesh size (MESH) of the landscape over time.

Compaction involves the formation of rounded patches in a circular shape that makes them more compact [53]. The more closely a patch shape is to a circle, the more it exhibits compaction. While a natural forest has a complex and irregular shape, basic geometry patch shapes show unnatural objects. Therefore, analysis of forest compaction enables us to assess disturbance in the forest using various shape metrics such as the shape index (SHAPE_MN, _AM) and circumscribing circle (CIRCLE_MN, _AM) at the class level. At the landscape level, area-weighted radius of gyration (GYRATE, _AM) is used to analyze compaction. Furthermore, GYRATE_AM also provides the overall characterization of the level of connectivity or subdivision of the landscape [53].

## 4. Results

### 4.1. Forest Type Classification

For the result of masking undesirable areas, we compared the predicted products with the reference data and evaluated them based on the index of overall accuracy (OA) for each year. The results obtained high accuracies for the images in 2007, 2020 and 2016 with an OA of over 0.87. The 2010 predicted image was the best with an OA of 0.99, followed by 2016 with 0.92 and 2007 with 0.87.

The behavior survey of only Landsat or only PolSAR data on forest objects does not show observable discrimination (Appendix C). For radar images, polarimetric parameters are not used to classify forest objects due to the saturation of entropy throughout the study area. Alpha and anisotropy display slight fluctuations on different forest objects, but they do not create good results in the discrimination. Nor does relying on the polarization of HH and HV signals provide better results. Therefore, with efforts to improve accuracy in forest classification, we have used combined data from optical and SAR data to extract information for forest types classification.

Another difficulty encountered in the classification process was the limited number of samples collected from the field, particularly in 2016 when only 106 samples were collected for the four forest types over the entire study area. The small number of samples was inadequate to develop a reliable classification algorithm based on the supervised method. To solve this problem, we used the semi-supervised classification with the addition of information from an unlabeled data source. However, to ensure the accuracy of the classification results, it was necessary to select an appropriate ratio between the number of labelled and unlabeled samples. The higher the percentage of unlabeled samples, the lower the accuracy [16]. To balance the number of unlabeled samples required and the classification accuracy, 200 random samples were created in the study area to ensure the ratio was approximately 60% for each year.

Overall, the classification accuracies were relatively high for 2007 with a kappa of 0.81 and OA of 0.86, respectively (Figure 4), while they were adequate for 2010 and 2016 with kappas of 0.76 and 0.74, respectively. The accuracies are generally the best for the rich forest over the entire time, with a user's accuracy of 100% in the years 2007 and 2010, and of 85.71% in 2016. This is followed by medium forest with over 75% in both user's and producer's accuracies, although sometimes it was misclassified as rich or poor forest. On the other hand, the classification accuracies were the lowest in 2010 for poor forest and in 2016 for restoration forest. The confusion matrix in 2010 and 2016 reveals a significant confusion between the poor and restoration forests, and they therefore cause the values of OA and kappa to be reduced at these times (Appendix D).

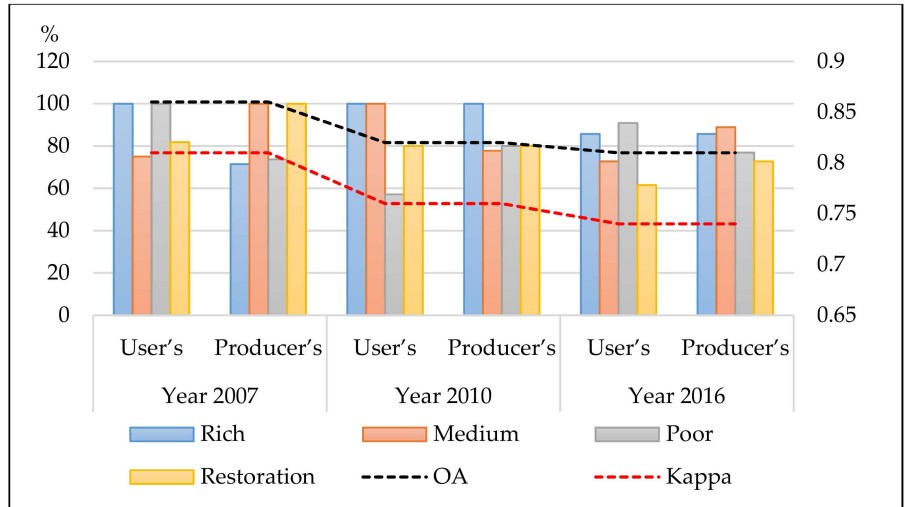

**Figure 4.** Forest types classification accuracies in user, producer (%), overall accuracy (OA), and kappa in the years 2007, 2010, and 2016.

## *4.2. Forest Pattern Analysis at the Class Level*

Based on the metrics of class area (CA) and percentage of landscape (PLAND_%), the natural forest of the study displayed a significant fluctuation within the nine years from 2007 to 2016 (Figure 5).

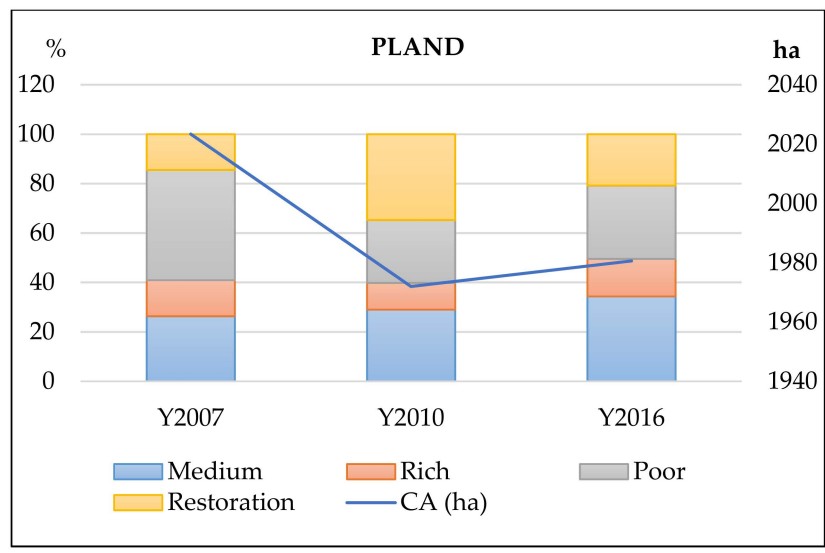

**Figure 5.** Variation of four forest types in the total class area of natural forest (CA_ha) and percentage of landscape (PLAND_%) for each forest type from 2007 to 2016.

In the period 2007–2010, CA decreased quickly with an average loss of 1713 ha per year. However, in the period 2010–2016, signs of recovery in CA appeared with an average gain of 144 ha each year. From 2007 to 2016, rich, medium, and restoration forests mainly demonstrated an increase, as shown by the gain of PLAND 1–8%, while PLAND showed reductions for poor forest of up to −15%. Furthermore, to assess the spatial variation of forest patterns, a set of parameters, comprising 11 metrics at the class level and 15 metrics at the landscape level, was selected based on evaluation of the indicators for universality, strength, and consistency. The selection method was based on factor analysis, clustering, and evaluation for the four different forest types at three different time points. Therefore, this set of metrics ensures the appropriateness and representativeness of forest structure analysis over time for this test site. The changes in each forest type, based on analyzing landscape metrics from 2007 to 2016, are shown in Table 4.

**Table 4.** Pattern metrics changes in the four forest types in class level metrics.

| Types | Metrics | % Change | | |
|---|---|---|---|---|
| | | 2007–2010 | 2010–2016 | 2007–2016 |
| Rich forest | SHAPE_MN | −20 | 18 | −5 |
| | CIRCLE_MN | −28 | 27 | −9 |
| | CIRCLE_AM | −14 | 19 | 2 |
| | CIRCLE_CV | 43 | −17 | 18 |
| | CONTIG_MN | −51 | 58 | −22 |
| | CONTIG_AM | −21 | 29 | 2 |
| | CLUMPY | −12 | 11 | −3 |
| | PLADJ | −15 | 16 | 1 |
| | COHESION | −12 | 12 | 1 |
| | AI | −15 | 16 | 1 |
| | nLSI | 67 | −38 | 4 |
| Medium forest | SHAPE_MN | −16 | 5 | −12 |
| | CIRCLE_MN | −3 | −3 | −6 |
| | CIRCLE_AM | 6 | −9 | −4 |
| | CIRCLE_CV | −6 | 6 | −1 |
| | CONTIG_MN | −11 | −1 | −11 |
| | CONTIG_AM | 9 | −5 | 4 |
| | CLUMPY | 7 | 1 | 9 |
| | PLADJ | 7 | −4 | 3 |
| | COHESION | 5 | −8 | −3 |
| | AI | 6 | −2 | 4 |
| | nLSI | −29 | 16 | −17 |
| Poor forest | SHAPE_MN | −9 | 6 | −4 |
| | CIRCLE_MN | −19 | 18 | −4 |
| | CIRCLE_AM | −6 | −6 | −11 |
| | CIRCLE_CV | 29 | −24 | −1 |
| | CONTIG_MN | −32 | 29 | −12 |
| | CONTIG_AM | −8 | −2 | −10 |
| | CLUMPY | −7 | 6 | −1 |
| | PLADJ | −6 | −2 | −8 |
| | COHESION | −2 | −6 | −8 |
| | AI | −7 | −1 | −7 |
| | nLSI | 53 | 4 | 60 |
| Restoration forest | SHAPE_MN | −8 | 3 | −6 |
| | CIRCLE_MN | −14 | 9 | −6 |
| | CIRCLE_AM | 9 | −7 | 2 |
| | CIRCLE_CV | 25 | −15 | 6 |
| | CONTIG_MN | −24 | 24 | −6 |
| | CONTIG_AM | 8 | −7 | 0 |
| | CLUMPY | −3 | −2 | −4 |
| | PLADJ | 6 | −5 | 1 |
| | COHESION | 5 | −6 | −1 |
| | AI | 5 | −4 | 1 |
| | nLSI | −25 | 33 | 0 |

In general, from 2007 to 2016, the forest types exhibited a relatively stable pattern with no significant changes in the group metrics of aggregation (AI, CLUMPY, PLADJ) but their pattern did show significant changes in patch shape structure (SHAPE, CIRCLE, CONTIG). In particular, rich, medium, and restoration forests had a low level of aggregation with the change percentage of AI and PLADJ ranging from just +1 to +4%. Conversely, the poor forest demonstrated an increased dispersion (AI −7%). However, this period was expressed by the moderate changes in shape with more compactness (SHAPE −4% to −12%) and contiguity (CONTIG_AM up to 4% excluding the poor

forest). The poor forest demonstrated the largest variation and had a trend of disaggregation (nLSI 60%) due to decreasing total area and percentage of the landscape. In summary, when evaluating the subperiods between 2007–2010 and 2010–2016, the forest types reflect an extreme fluctuation and totally different behavior.

### 4.2.1. Period 2007–2010

This period expressed a growth in the percentage of landscape occupied by medium and restoration forests, as well as a decline in rich and poor forests. Therefore, they exhibit completely different processes in spatial fluctuations (Table 4).

Rich forest displayed a moderate decrease (PLAND −4%) and strong disaggregation in this period. The disaggregation is reflected in a decrease in AI −15%, CLUMPY −12%, and PLADJ −15%, and an increase in nLSI (+67%). The patterns show more compactness and less physical connectivity, as a result of reducing complexity in geometric shape (SHAPE_MN −20% and CIRCLE_AM −14%), decreasing contiguity, and continuity (CONTIG_MN −51% and COHESION −12%). The related circumscribing circle coefficient of variation (CIRCLE_CV) with a high value indicates the various changes in patch shapes for rich forest.

Similar to rich forest, poor forest exhibited slightly increased dispersion corresponding to a decrease in clumpiness and aggregation (–7% for both the change of CLUMPY and AI). This is due to shrinkage in the percentage of landscape (PLAND −19%) and the disappearance of like adjacencies in the same patch type (PLADJ −6%). It also coincides with the tendency to increase compactness (SHAPE_MN −9%).

The medium and restoration forests had growth in terms of area (PLAND 3% and 20%, respectively) and demonstrated a different process than rich and poor forest. The patterns display a moderate aggregation, higher connectivity, and compactness. In addition, the growth in area, together with the drop in contiguity index (CONTIG_MN -3% and −24% for the medium and restoration, respectively), reflect the process of creating larger patches from the clumpiness of small adjacencies and the distribution scattered in the landscape.

### 4.2.2. Period 2010–2016

In this period, rich forest performed more aggregation than other forest types. The appearance of new patches (PLADJ +16%) resulted in an increase in spatial connectedness (CONTIG_MN +58%) and improved the continuity of this class in the landscape (COHESION +12%). This also meant a gain in the aggregation process (AI +16%, CLUMPY +11%, and nLSI –38%). The growth in PLAND coincided with a higher area-weighted mean contiguity of each patch (CONTIG_AM +29%), indicating the appearance of larger patch sizes.

Medium and poor forest demonstrated less area variation than in the previous period with a slight increase. However, there was a negligible decrease in tendency of aggregation (AI –1 to –2%), continuity (COHESION –6 to –8%), and connectedness (CONTIG_AM –2% to –5%) for both types. In poor forest, there was a different tendency of the mean index and area-weighted mean index in CIRCLE and CONTIG due to measuring the patch-centric and landscape-centric perspectives. The increase in the related circumscribing circle shows a trend of elongation based on evaluating entire patches (CIRCLE _MN +18%), but displays the opposite trend based on evaluating an arbitrary patch selected randomly from the landscape (CIRCLE_AM –6%).

Similarly, CONTIG_MN demonstrated a significant increase (+24%) and performed a higher level of spatial connectedness in poor forest. However, the drop in CONTIG_AM (–2%) together with the expansion of area partly revealed the presence of new small patches.

An extreme decline in the restoration area was recorded during this period, resulting in increasing dispersion (nLSI +33%), a higher level of complexity in shape structure (SHAPE_MN +3% and CIRCLE_MN +9%), and less contiguity (CONTIG_AM –7%).

### 4.3. Forest Pattern Analysis in Landscape Level

This period was marked by a rapid decrease in the total landscape area of natural forest, from 202,300 ha in 2007 to 197,200 ha in 2010, followed by a slight increase to 198,100 ha in 2016. This caused a reduction in the percentage of the landscape and a sharp decline in patch size distribution (AREA_AM –60%) (Table 5). In addition, there was a decrease in symmetry in the patch distribution in the landscape (IJI –8%). The moderate decline in SHAPE (–20%) and GYRATE (–26%) demonstrated more compactness and less complexity in spatial patterns. The continuity and connectedness of the forest pattern also tended to decrease (CONTIG_MN –21%)). In general, the natural forest experiences increased fragmentation over the entire landscape, which involved an increase in landscape area with shrinkage of patch size and disproportionate distribution of patches.

**Table 5.** Pattern metrics changes in landscape level metrics.

| Metrics | % Change | | |
|---|---|---|---|
| | **2007–2010** | **2010–2016** | **2007–2016** |
| AREA_AM | −49 | −22 | −60 |
| GYRATE_AM | −7 | −20 | −26 |
| SHAPE_AM | 7 | −25 | −20 |
| CONTIG_MN | −30 | 14 | −21 |
| IJI | −5 | −2 | −8 |

## 5. Discussion

To assess the trend of natural forest changes in the study area, we compared the results with those in global and tropical regions, as well as in Vietnam overall, in the same period. Keenan et al. [3] reviewed the dynamics of global forest area between 1990 and 2015 based on statistics from the FAO global forest resources assessment 2015. Worldwide, the natural forest area declined by 2% between 2005 and 2015, with the vast majority of the losses occurring in the tropics where the rate of loss fell by 7.2 million ha.y$^{-1}$. Compared to the trend of forest transition worldwide and in Vietnam overall, the status of forest loss in the study area is similar in the period from 2007–2010. This status is confirmed by the findings of Quy Van Khuc et al. [54] that degradation mainly occurred in natural forest at the rate of 3–4%. Cochard's [55] review of studies also demonstrated a slow increase in natural forest in the period 2000–2013 in Thua Thien Hue province. From 2010–2016, the natural forest in the study area demonstrated the opposite trend. While there was a significant decrease in the natural forest worldwide and in the tropics generally, growth occurred in Vietnam and in the study area. Due to the shortage of previous studies, it was only possible to compare the general trend of natural forests. It is difficult to compare fluctuations in forest types of rich, medium, poor, and restoration types because there are few documented records for the study area in particular, and Vietnam in general, particularly in the period 2010–2016. Therefore, the findings of this study contribute to the understanding of the transition of natural forest types in recent years, particularly in the ecological processes in terms of spatial patterns that have still not received adequate attention.

Analysis of the reflectance behavior on some bands on Landsat and backscatter on SAR images (Figure A1) demonstrates on histograms the overlap of all four forest types. In image data from 2007 and 2010, rich forest exhibited better distinctions than other forest types. Histogram analysis of forest types in 2016 shows little separation, so its accuracy was lower than that of 2007 and 2010. This low separation is due to the characteristic of natural forest, with its combination of various canopy stories and species diversity. Sparser wood trees have more vines, which cover the whole canopy. Therefore, it is difficult to detect the difference between forest types based on optical images. Despite having a long wavelength that can supposedly penetrate the canopy and reach the trunk, L-band signals still demonstrate a low difference between polarization signals. In this study, to enhance the differences

between classes, a multivariate model was essential to observe objects under multidimensional space and provide more information and attributes for objects.

The change of certain forest types between any two periods comprises the net effect [3] of conversion from any one forest type to another or non-forested area and natural regeneration or restoration. A conversion matrix was used to clearly illustrate transition areas between forest types in this study area in the period 2007–2016 (Table 6). In this table, the cross cells demonstrated no change values in terms of percentage of forest types' area. The rows demonstrate an increase in the proportion converted from other types. The columns demonstrate a decrease in the proportion converted into other types. The net area change is the total effect of increase and decrease in the area of specific forest types.

**Table 6.** Conversion matrix of forest types between 2007 and 2016 in percentage (%) and area (ha).

| 2007 \ 2016 | Rich | Medium | Poor | Restoration | Others | Area Increase (ha) |
|---|---|---|---|---|---|---|
| Rich | 19 | 16 | 13 | 1 | 1 | 24,569 |
| Medium | 36 | 29 | 30 | 13 | 4 | 52,749 |
| Poor | 21 | 23 | 26 | 29 | 3 | 34,970 |
| Restoration | 14 | 18 | 15 | 26 | 2 | 33,585 |
| Others | 11 | 14 | 15 | 31 | 90 | |
| Area decrease (ha) | −24,046 | −37,955 | −66,409 | −21,732 | | |
| **Net area change (ha)** | **522** | **14794** | **−29,437** | **11,853** | | |

From the conversion matrix of forest types between 2007 and 2016 in this study area, we considered three main findings. First, the net trend of natural forest comprised a small loss of area, but this was due to two opposite trends of an area increase in one place and a decrease elsewhere. Second, the levels of forest restoration and deforestation were nearly equal (total of 145,873 ha and 150,143 ha, respectively) and occurred simultaneously in the study area during this period. Third, there was a strong internal transition between forest types and an external transition between them and other land use/land cover types. Medium forest had the highest gain area, followed by restoration forest, at 14,795 ha and 11,852 ha, respectively. Poor forest showed a sharp loss, while rich forest had an adequate increase. When considering the percentage of conversion area, the most dramatic transformation was in rich forest, which changed to medium forest at a rate of 36%, but was compensated by medium (16%) and poor forests (13%). However, when considering the changing area, medium and poor forest had areas of both high increase and decrease. Changes from natural forest to other types were the strongest in restoration forest at 31% of its area. Restoration forest is the most vulnerable forest type because it is often distributed in areas that are easily accessible and affected by human and agricultural activities.

The spatial changes in natural forest types presented in Figure 6 showed two change directions: increase and decrease. The area loss of forest types occurred throughout the study area, but it was mainly distributed near water bodies such as rivers and streams. The local population distribution is often concentrated in the downstream of rivers where conditions for agriculture are developing. Therefore, natural forests near rivers are easily deforested and degraded due to human activity. In the other direction, the expansion of rich forest created larger fragments and scattered distribution in the study area, resulting in increasing compactness, less connectivity, and higher isolation. The expansion in other types occurred more evenly and therefore with greater connectivity.

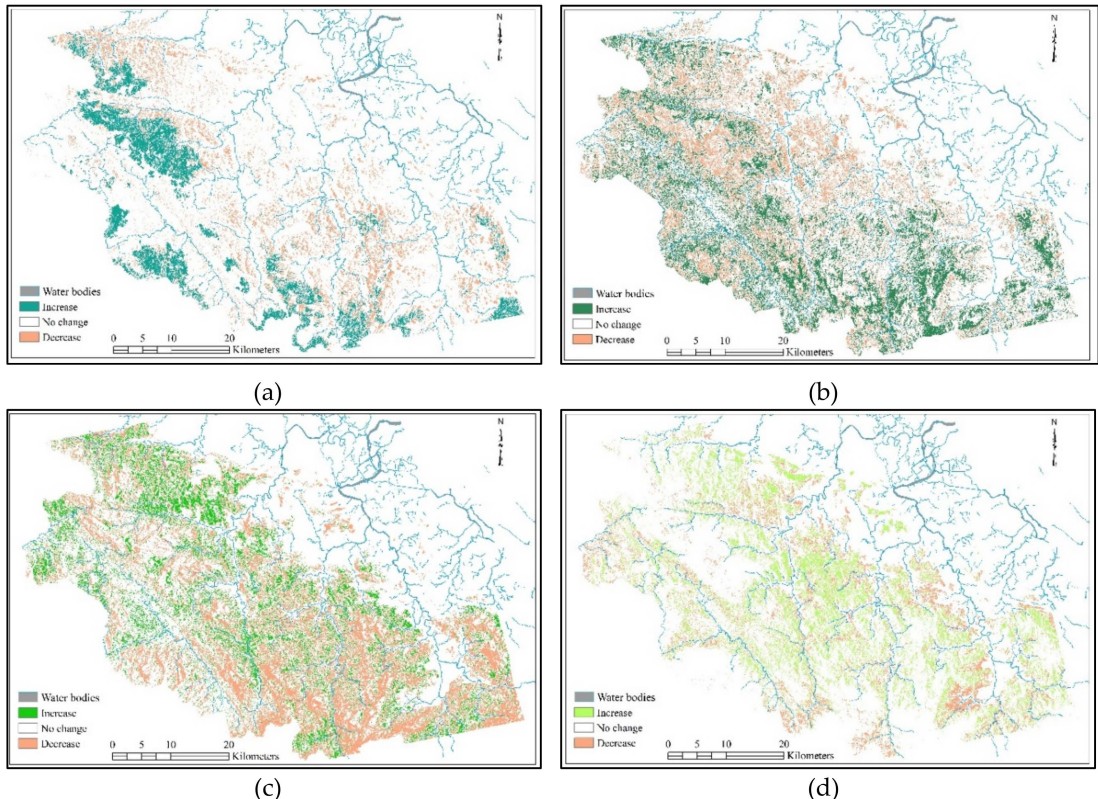

**Figure 6.** Changes in (**a**) rich forest, (**b**) medium forest, (**c**) poor forest, and (**d**) restoration forest between 2007 and 2016.

During this period, there were many factors affecting forest dynamics. The policies of prohibiting logging in natural forests and enhancing forest protection and restoration are considered to be the correct policies in terms of reducing natural forest degradation, which was implemented by the Vietnamese government since the early 1990s [56]. However, illegal logging still occurred [57] due to the increasing demand for wood from population pressure, which is the main reason for the continued decline of natural forests in the period 2005–2010. There were also many other causes, such as poverty, forest resources, population density, agricultural production, and province-level governance [54]. In parallel with the logging ban policy in natural forest, Vietnam has successfully socialized forestry organization, calling for public participation in afforestation and forest protection, and resulting in reduced deforestation and degradation and improved long-term income for people in rural mountainous areas. The speed of loss of natural forests has also decreased slightly and there have been signs of increase from 2010 to the present day. In 2016, Vietnam began to introduce bans on natural forest wood exploitation into the law on forest protection and development, which is the most powerful law in forestry. Simultaneously, it maximized the closure of natural forests, did not convert natural forests to other purposes, and did not convert poor natural forests to industrial crops. This is the driving force behind reductions in degradation and prevention of illegal logging, and allows us to predict recovery and increase in the quality of natural forests in the future.

Generally, this study provides information on the dynamics and spatial processes of natural forest change in a given study site between 2007 and 2016. The result obtained demonstrates the general trend of forest types conversion and provides useful information for sustainable forest planning.

## 6. Conclusions

There is an essential requirement for forest management and protection to classify natural forests and assess their fluctuations over time. However, classifying the natural forest types in tropical areas using remote sensing images is challenging because of the very similar information captured by

remotely sensed data as well as the constraint of samples data. Furthermore, there is a lack of research assessing forest transition in the natural forest from the perspective of landscape ecology, which can be used for forest structure management, and to quantitatively characterize the spatial patterns of forest landscapes. In this study, we addressed these issues by applying semi-supervised classification for data integration of optical and SAR data.

The combination of Landsat and PolSAR data resulted in improved discrimination of forest types. The using of multi-source remotely sensed data can provide more information about the object, as well mitigate the disadvantages of Landsat images (cloud, lower spatial resolution), and limited information regarding objects in PALSAR/PALSAR-2 image (only two polarization HH and HV).

In this study, we assessed the potential of a proposed semi-supervised model developed and validated for mapping forest types and assessed the process of forest transition in a tropical natural forest in Vietnam. The model produced high accuracies in the classified images in 2007, 2010, and 2016 with over 0.74 for kappa, and over 0.8 for OA. Additionally, landscape metrics were used to evaluate the forest changes based on the spatial processes, such as aggregation, fragmentation, and compaction. At the class level, the poor forest demonstrated the largest variation with more dispersed growth patterns, while other types had a low level of aggregation. At the landscape level, the natural forest experiences increased fragmentation, which involved an increase in landscape area with shrinkage of patch size and disproportionate distribution of patches.

We recommend that future research include comparison of different models to estimate the improvement resulting from the proposed model. Another important study that should be conducted is testing of the proposed methods on larger areas.

**Author Contributions:** Conceptualization, T.T.C.T.; methodology, T.T.C.T.; validation, T.T.C.T., H.T.; formal analysis, T.T.C.T.; investigation, T.T.C.T.; resources, H.T., N.Q.T.; data curation, T.T.C.T.; writing—original draft preparation, T.T.C.T.; writing—review and editing, T.T.C.T., H.T.; visualization, T.T.C.T.; supervision, H.T., X.W.; project administration, T.T.C.T.; funding acquisition, H.T.

**Funding:** This research received no external funding.

**Conflicts of Interest:** The authors declare no conflict of interest.

## Appendix A.

**Table A1.** List of 56 metrics in class level (C) and 63 metrics in landscape level (L).

| Number | Variable | | Description |
|---|---|---|---|
| 1 | CA | C | Total class area |
| 2 | CLUMPY | C | Clumpiness index |
| 3 | CPLAND | C | Core area percentage of landscape |
| 4 | NLSI | C | Normalized landscape shape index |
| 5 | AI | C, L | Aggregation index |
| 6 | AREA_AM | C, L | Area-weighted mean patch size |
| 7 | AREA_CV | C, L | Patch size coefficient of variation |
| 8 | AREA_MN | C, L | Mean patch size |
| 9 | CAI_AM | C, L | Area-weighted mean core area index |
| 10 | CAI_CV | C, L | Core area coefficient of variation |
| 11 | CAI_MN | C, L | Mean core area index |
| 12 | CIRCLE_AM | C, L | Area-weighted mean circumscribing circle |
| 13 | CIRCLE_CV | C, L | Circumscribing circle coefficient of variation |
| 14 | CIRCLE_MN | C, L | Mean coefficient of variation |
| 15 | COHESION | C, L | Patch cohesion |
| 16 | CONNECT | C, L | Connectance index |
| 17 | CONTIG_AM | C, L | Area-weighted contiguity index |
| 18 | CONTIG_CV | C, L | Contiguity index coefficient of variation |
| 19 | CONTIG_MN | C, L | Mean coefficient of variation |
| 20 | CORE_AM | C, L | Area-weighted mean core area |

**Table A1.** *Cont.*

| Number | Variable | | Description |
|--------|----------|------|-------------|
| 21 | CORE_CV | C, L | Core area coefficient of variation |
| 22 | CORE_MN | C, L | Mean core area |
| 23 | DCAD | C, L | Disjunct core area density |
| 24 | DCORE_AM | C, L | Area-weighted mean disjunct core area |
| 25 | DCORE_CV | C, L | Disjunct core area coefficient of variation |
| 26 | DCORE_MN | C, L | Mean disjunct core area |
| 27 | DIVISION | C, L | Division index |
| 28 | ED | C, L | Edge density |
| 29 | ENN_AM | C, L | Area-weighted mean nearest neighbor distance |
| 30 | ENN_CV | C, L | Nearest neighbor distance coefficient of variation |
| 31 | ENN_MN | C, L | Mean nearest neighbor distance |
| 32 | FRAC_AM | C, L | Area-weighted mean fractal dimension |
| 33 | FRAC_CV | C, L | Fractal dimension coefficient of variation |
| 34 | FRAC_MN | C, L | Mean fractal dimension |
| 35 | GYRATE_AM | C, L | Mean radius of gyration |
| 36 | GYRATE_CV | C, L | Radius of gyration coefficient of variation |
| 37 | GYRATE_MN | C, L | Mean radius of gyration |
| 38 | IJI | C, L | Interspersion/juxtaposition index |
| 39 | LPI | C, L | Largest patch index |
| 40 | LSI | C, L | Landscape shape index |
| 41 | MESH | C, L | Mesh index |
| 42 | NDCA | C, L | Number of disjunct core areas |
| 43 | NP | C, L | Number of patches |
| 44 | PAFRAC | C, L | Perimeter–area fractal dimension |
| 45 | PARA_AM | C, L | Area-weighted mean perimeter–area ratio |
| 46 | PARA_CV | C, L | Perimeter–area ratio coefficient of variation |
| 47 | PARA_MN | C, L | Mean perimeter-area ratio |
| 48 | PD | C, L | Patch density |
| 49 | PLADJ | C, L | Proportion of like adjacencies |
| 50 | PLAND | C, L | Proportion of landscape |
| 51 | SHAPE_AM | C, L | Area-weighted mean shape index |
| 52 | SHAPE_CV | C, L | Shape index coefficient of variation |
| 53 | SHAPE_MN | C, L | Mean shape index |
| 54 | SPLIT | C, L | Splitting index |
| 55 | TCA | C, L | Total core area |
| 56 | TE | C, L | Total edge |
| 57 | CONTAG | L | Contagion |
| 58 | MSIDI | L | Modified Simpson's diversity index |
| 59 | MSIEI | L | Modified Simpson's evenness index |
| 60 | PR | L | Patch richness |
| 61 | PRD | L | Patch richness density |
| 62 | RPR | L | Relative patch richness |
| 63 | SHDI | L | Shannon's diversity index |
| 64 | SHEI | L | Shannon's evenness index |
| 65 | SIDI | L | Simpson's patch density |
| 66 | SIEI | L | Simpson's patch evenness |
| 67 | TA | L | Total landscape area |

## Appendix B.

**Table A2.** Universality, strength, and consistency at class level.

| Cluster | Members | % Total | Eigenvalue | % Variation Explained | Average in Group Correlation |
|---------|---------|---------|------------|----------------------|------------------------------|
| 1 | 9 | 96 | 4.01 | 7.29 | 0.81 |
| 2 | 4 | 83 | 2.13 | 3.87 | 0.97 |
| 3 | 6 | 93 | 2.35 | 4.27 | 0.71 |
| 4 | 8 | 100 | 4.05 | 7.36 | 0.92 |
| 5 | 4 | 75 | 1.88 | 3.43 | 0.85 |
| 6 | 3 | 92 | 1.00 | 1.83 | 0.61 |
| 7 | 2 | 83 | 0.71 | 1.29 | 0.65 |
| 8 | 7 | 89 | 2.62 | 4.77 | 0.68 |
| 9 | 3 | 100 | 1.22 | 2.22 | 0.74 |
| 10 | 4 | 83 | 1.69 | 3.07 | 0.76 |
| 11 | 5 | 85 | 2.15 | 3.91 | 0.78 |

**Table A3.** Universality, strength, and consistency at the landscape level.

| Cluster | Members | % Total | Eigenvalue | % Variation Explained | Average in Group Correlation |
|---------|---------|---------|------------|----------------------|------------------------------|
| 1 | 10 | 100 | 5.27 | 8.50 | 0.85 |
| 2 | 7 | 100 | 4.17 | 6.72 | 0.96 |
| 3 | 10 | 100 | 4.85 | 7.83 | 0.78 |
| 4 | 8 | 100 | 3.55 | 5.73 | 0.72 |
| 5 | 8 | 100 | 4.03 | 6.49 | 0.81 |
| 6 | 5 | 100 | 2.13 | 3.43 | 0.69 |
| 7 | 2 | 100 | 1.22 | 1.98 | 0.99 |
| 8 | 2 | 100 | 1.24 | 2.00 | 1.00 |
| 9 | 8 | 100 | 3.64 | 5.87 | 0.73 |
| 10 | 2 | 100 | 1.18 | 1.91 | 0.95 |

**Appendix C.**

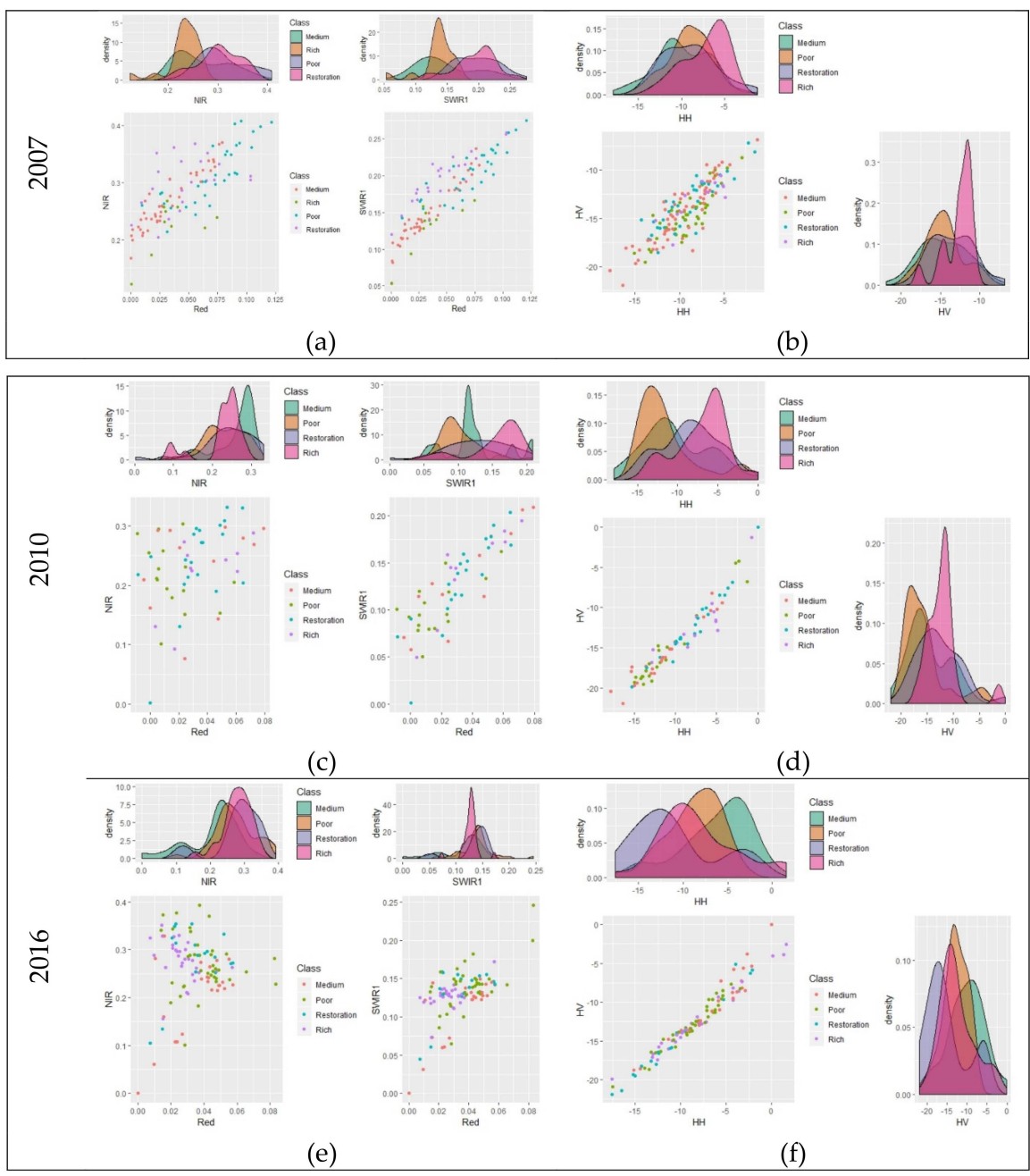

**Figure A1.** Distribution and density of some parameters (HH and HV signals in decibels for SAR data, and red, near-infrared, and shortwave infrared 1 in reflectance for Landsat data) in four forest types in three years 2007 (**a**,**b**), 2010 (**c**,**d**), and 2016 (**e**,**f**).

## Appendix D.

**Table A4.** Confusion matrix of classification in 2007.

| Prediction | Medium | Rich | Poor | Restoration | User's |
|---|---|---|---|---|---|
| Medium | 15 | 1 | 4 | 0 | 75.00 |
| Rich | 0 | 5 | 0 | 0 | 100.00 |
| Poor | 0 | 0 | 14 | 0 | 100.00 |
| Restoration | 0 | 1 | 1 | 9 | 81.82 |
| Producer's | 100.00 | 71.43 | 73.68 | 100.00 | |
| Overall accuracy | | | 0.86 | | |
| Kappa | | | 0.81 | | |

**Table A5.** Confusion matrix of classification in 2010.

| Prediction | Medium | Rich | Poor | Restoration | User's |
|---|---|---|---|---|---|
| Medium | 7 | 0 | 0 | 0 | 100.00 |
| Rich | 0 | 4 | 0 | 0 | 100.00 |
| Poor | 2 | 0 | 4 | 1 | 57.14 |
| Restoration | 0 | 0 | 1 | 4 | 80.00 |
| Producer's | 77.78 | 100.00 | 80.00 | 80.00 | |
| Overall accuracy | | | 0.82 | | |
| Kappa | | | 0.76 | | |

**Table A6.** Confusion matrix of classification in 2016.

| Prediction | Medium | Rich | Poor | Restoration | User's |
|---|---|---|---|---|---|
| Medium | 7 | 0 | 2 | 1 | 70.00 |
| Rich | 0 | 12 | 2 | 0 | 85.71 |
| Poor | 0 | 0 | 12 | 1 | 92.31 |
| Restoration | 1 | 2 | 0 | 5 | 62.50 |
| Producer's | 87.50 | 85.71 | 75.00 | 71.43 | |
| Overall accuracy | | | 0.81 | | |
| Kappa | | | 0.74 | | |

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
