# Peer review of "Semi-Supervised Classification and Landscape Metrics for Mapping and Spatial Pattern Change Analysis of Tropical Forest Types in Thua Thien Hue Province, Vietnam"

_forests, doi:10.3390/f10080673_

Round 1

Reviewer 1 Report

Interesting study, with high potential for improvements. The value of the manuscript would improve, if the authors make the remote sensing research related component stronger. Without that, the study looks like a technical report describing forests in one province of Vietnam, nevertheless, there are all chances to get good paper on developing or validating methodological approaches to get the information on the tropical forests, which is validated in the study area. Then, information, achieved using the proposed solution, is further used for forestry related interpretations.

Nevertheless, there are many chances to improve the manuscript. Please, find more detailed comments below:

Abstract (comments formulated before reading the body text of the manuscript):

Line 14: You leave the “Research Highlights” empty, however, research contribution of your study could be highlighted here. E.g. your conclusion on your methodological approach could fit here. Otherwise, also looking at your objectives, one may expect to find more technical report than research paper.

Line 15: What do you understand as “Natural forest”, how long is “long-term”, why year 2005 is so special, do you have the information on the period before 2005? As for the abstract, you could try find some more convincing formulation, “lack of information on the changes and the role of remote sensing with specific methodological issues…”.

Line 18: Well, it is always difficult to classify, but there are usually reasons for that?

Line 19: Why your area is specific – would the classification be easy in other areas? What do you mean under “limited number of samples”? One may easily say – make a larger sample.

Line 20: Better avoid using “these problems”, as, in fact, you have not formulated scientific problem yet. Objective should be better seen, a kind of “… validate/develop using of (remote sensing/specific methodological approach) to … map changes and ...”

Line 21: You should not be too specific here – you state “between 2007 and 2016”. Would the methodological approaches and the trends in results be different for other period?

Lines 22-28: Material and methods – the first sentence better fits the research highlights or objectives. Here, you should precisely identify the study object, data, classification and validation algorithms used.

Lines 27-28: This information refers to the results.

Lines 29-32: Results. You should introduce the results more carefully. Here you state about “increasing and decreasing forest”, but the reader is still not aware about the actual extents, i.e. the facts behind your statements. Any methodological finding?

Lines 33-35: you should conclude on your findings, good or bad your approach, could it be upscaled to other regions. You have not introduced the role of your approach in conservation and planning, thus, why to use this in conclusion.

Lines 36-37: I miss remote sensing or something like that among the keyword. Province and country name should not be used as the keywords, as you have them in the title anyway.

Introduction. Could be easily improved, especially providing more thorough characterization of image processing algorithms and state-of-the-art of solutions to map/classify vegetation types in tropical forests, shorth scientific introduction of the landscape metrics.

Lines 41-53: Somehow the style of this paragraph could be improved. E.g. you state “Such research can be …” – but do you focus here on the research. What was the motivation for the new approach? I.e. what was the previous approach and why the Kull’s approach was needed and how it relates to your study. I can hardly agree that “landscape ecology” as such delivers only information on forest cover and characteristics. Do you automatically assume remote sensing as forest inventories?

Line 53: It is to challenging to aim “to provide more information” with scientific paper. Better call the need for more information as an overall challenge.

Lines 54-55: Somehow you avoid the “forest inventory”, but maybe there are reasons for that. You state “The characteristics of natural forests in the tropics make their classification using remote… “ – do you classify the characteristics or remotely sensed images?

Line 57: Hardly suitable term “biodiversity” to identify, I suppose, large diversity of tree species.

Line 58: Theoretically, the more species, the more spectral properties you have. I suppose, the issue is in number of species and relatively similar reflectance properties?

Line 59: I can hardly agree that radar or lidar have contributed significantly to identification of forest types (if you consider the types compatible with tree species).

Lines 60-61: What do you understand under multi-data. Do you plan to focus on multi-source remote sensing data? You could emphasize multi-source approach, maybe even including among your research challenges and/or research objectives. Nevertheless, you would need to elaborate more on this topic, also using more references.

Lines 64-65: I suppose, the term “supervised algorithm” etc. is used in context of image classification. You could make clearer introduction of the semi-supervised (image) classification, also using some references. Here, your references from 14 to 18 are to image classification techniques (listing them), rather than discussing on the issues to apply them. How could you use e.g. deep learning with insufficient training data? Also, I suppose, that the title “semi-supervised” should not be used without additional context (image classification?). Also, make sure, that you use it in a uniform way. E.g. you have “Semi-supervised model” in the title, “semi-supervised algorithm”, “semi-supervised classification approach” and “semi-supervised method” in the Introduction.

Line 77: Define the self-learning approach, maybe identify it as one of objectives of your study.

Lines 83-89: Somehow I would find a place for such paragraph above the introduction of remote sensing tasks. … a valuable tool, … however, we need data… which is available through remote sensing. However, the use of remote sensing is not too easy due to…  Ok, it seems that you are doing dome forest change pattern analysis in your study. Then this should be more supported with the description of the issue in context of your and, especialy, other studies. It is clearly not enough to state that “… landscape metrics are a very valuable tool …”.

Lines 103-111: Formulate clearly the objectives of your study in this paragraph, maybe even numbering them. If you state, that the objective is to assess the changes of forest types in very specific region, then we are dealing with technical report, potentially not interesting neither for international audience, nor the scientific journal. However, it is clear, that you are going to deal with significant methodological issues (multi-source, semi-supervised, pattern analysis), so, why not to put the real research contribution first, saying that this development/validation is done through assessing “… forest type changes…”.

Study area

Line 115: By the way, how were the figures on decreasing forest area achieved? Do you think this is a lot?

Lines 120-124: The text here hardly refers to description of the study area.

Line 125: As you state, that you “classified”, then this reminds already Methodology. Also – I would say, that you should introduce the forest types in the introduction. The reader should understand immediately, what is your understanding of forest type. E.g. I thought the talk will be about tree species related forest types. Then ok regarding Lidar, but spectral properties not very relevant?

Line 136: Do not find where the Figure 1 is introduced in the text. As there are satellite image scenes identified, I suppose you also introduce here the materials used? Unfortunately, the quality of Figure 1 is unreadable to provide more comments. Legend to part d? Global geographic context?

Data and methods: The major issue with this chapter is that a lot of text is used to explain the methods used. I.e. the details are provided, together with numerous references, discussion of suitability of one or another approach. The suggestion is to concentrate on the methods used by the authors to conduct their study at the level, other potential researcher could repeat the study. Materials, overlapping with the Introduction or Discussion, could be moved to appropriate chapter, especially assuming that e.g. the Introduction is rather weak in terms of review of already known scientific facts and elaboration on the contribution by the authors.

Line 165: What do you assume under “unlabelled sample”?

Line 172: It seems that you put the figures in the text before they are first mentioned?

Lines 181-186: This sounds more like a discussion, i.e. not relevant to the chapter of Material and Methods. Did you calculate the NDVI? Did not understand (probably the style issue) the following – “… EVI should be removed from the topographic effect before calculation”. Topographic effect or EVI? Did you apply the topographic normalization, finally?

Line 204: PolSAR?

Line 224: Do not you classify the forest into 4 types? Or the sub-chapter is about classifying the remotely sensed data?

Line 235: Maybe it is better to refer to your objectives talking about the object of the study?

Lines 244-249: Here you also seem to discuss the suitability of specific methods. I would suggest just identify the methods and leave supporting information either for Introduction, or the Discussion

Lines 262-271: The same issue as above. Maybe you could consider to use sub-chapter for methodological underpinning? In the Methods (or Annexes) one would expect detailed settings of the algorithms used

Line 297: Style (?): Each image calculated 56 metrics at the class level

Lines 302-303: I suppose, the first sentence is redundant or the style should be changed “to…, we did…”

Lines 306-310, 326-334: Here, as in many places above, you introduce the methods, but not what have you done. I.e. this information is redundant for Material and Methods used to conduct the study

Line 313: Grouped?

Results and Discussion: the overlap with other chapters of the paper. Authors should avoid the explaining what they have done (this is described in the previous chapter), but focus on the results, i.e. what has been estimated.

Lines 350-352, 393-396: Overlap with Methods

Line 354: Improve the style – “The results obtained were highly accurate for all years with an OA of over 0.87”. The accuracy of classification was usually high (?)

Line 355: “The non-forest class was then masked out” – we do already know this from the Methods, as many other facts you repeat later.

Line 358: style – what is the forest image? In fact, are not you working with the masked-out parts of satellite images? But this text overlaps with the methods

Line 360: Do not understand “For optical images, layers of natural forest observe similarities in shape”

Line 379: Nice pictures, though of little scientific value.

Line 389: Where are the error matrices? Provide them e.g. in Supplementary Materials, I suppose would be very interesting to look at, also some potential question (not asked by the reviewer) on the validation metrics (OA & Kappa) would disappear

Line 447: Wrong numbering

Line 480: Discussion – consider using it as a separate subchapter at the same level with Results and Conclusions. Some “discussion” could fit to the Results. However, the main issue here seems to be associated that the authors focus to much on “what they have discovered” practically forgetting about “what they have done and how”. I would say the comparison with national data is redundant. Did I miss the implications? Maybe the authors could suggest their approach for official statistics and discuss then the differences in approaches and their advantages/disadvantages? In the conclusions, the authors state that they “addressed … difficulties by applying semi-supervised classification for data integration of optical and SAR data”. However, in the Discussion there is just a statement, that “…easy to map using satellite data [3], but difficult to classify into subclasses due to its heterogeneity”. Are there any other studies confirming this? Or, what approaches should be used instead? Even though authors conclude, that “study confirms the potential of the semi-supervised method and landscape metrics on solving some constraints in using remote sensing for natural forest resources management”, I miss this in the Discussion, highlighting the importance of findings by the authors in context of other research.

Conclusion

Line 606: Style: It is some challenges to classify.

Lines 616-617: Style: “We found that two opposite trends, of increasing and decreasing, were occurring at nearly the same level and simultaneously in the study area during this period” – would you understand what was in the mind here? This would be difficult to understand even having neighbouring sentences provided. Check the style very carefully, especially in the Conclusions. Make sure, that you make compatible the conclusions with the Abstract and objectives in the Introduction. Avoid unnecessary sentences which introduce only uncertainty in your statements. Focus on the message you want to deliver, what have you discovered first and leave the interpretations for the reader.

Reviewer 2 Report

The research presented in this paper regards an interesting field. Overall, the paper is very good, with clear presentation of the study developed.

Moreover, this manuscript provides an interesting contribution to the existing literature, which could enrich the knowledge on forest transition and strategic planning for sustainable nature conservation.

Therefore, I advise the manuscript for publication after some minor issues.

My remarks are following below:

 -  In the "Conclusion" section, the authors should discuss the limitations of their research.

 -   The presentation of the figures could be improved.

 -    Please check the paragraph numbering.

Round 2

Reviewer 1 Report

I would like to acknowledge the improvements made by the authors. The review would be easier if the authors had provided more detailed summary on the changes made in their responses. Nevertheless, I consider the manuscript has been significantly improved and is acceptable for publishing. Some minor comments are provided below. Also, the authors consider checking the style – in general, it is ok, but I feel there is some space for improvement.

Point 2: The question regarding the “natural forest” was not because the reviewer was wandering on the interpretation of the term. E.g. the term “normal forest” may also be associated with classical German forestry concept originating from the 19th century. I would suggest making short statement in the text first using this term, something like “… normal forest, based on the FAO definition [include reference] …”.

Point 5. Current version looks fine. Once again on the reasons for this comment - the difficulties with the sample size are obvious, and there is no need to emphasize them in abstract, unless you suggest something special to deal with.

Point 12: Do not you think PALSAR and PALSAR-2 are redundant?

Point 23: The objectives of the study are better structured, however, still not clear enough. E.g. in the lines 433-438 (using pdf version with the changes tracked) – the 1st objective is to assess the potential of …, however, next you state that you aim to develop this model (is it not available yet?), maybe use improve this approach (or something like that, as you do already have it). I would also suggest to limit the use the term “normal forest” here – does not your approach apply forests in general? The sentence “This study contributes …” is vague, by the way, like is your statement on “strategic planning” – if not focusing on this issue in more details. Also, do not use “objective” in line 420, better consider this paragraph as a research challenge.

I would still say, that the authors could be more precise with the Conclusions. Both in terms of style and information contents used. Better use wording on what you have discovered, rather than assessed. Also, did the model “achieve”? You could be more specific in e.g. “… provided much information…” in Conclusions.
